# Co-Expression Networks for Causal Gene Identification Based on RNA-Seq Data of *Corynebacterium pseudotuberculosis*

**DOI:** 10.3390/genes11070794

**Published:** 2020-07-14

**Authors:** Edian F. Franco, Pratip Rana, Ana Lidia Queiroz Cavalcante, Artur Luiz da Silva, Anne Cybelle Pinto Gomide, Adriana R. Carneiro Folador, Vasco Azevedo, Preetam Ghosh, Rommel T. J. Ramos

**Affiliations:** 1Institute of Biological Sciences, Federal University of Para, Belem 66075-110, PA, Brazil; edianfranco@ufpa.br (E.F.F.); cavalcante.analidia7@gmail.com (A.L.Q.C.); arturluizdasilva@gmail.com (A.L.d.S.); carneiroar@gmail.com (A.R.C.F.); 2Biological Engineering Laboratory, Science and Technology Park Guama, Belem 66075-750, PA, Brazil; 3Instituto Tecnológico de Santo Domingo (INTEC), Santo Domingo 10602, DN, Dominican Republic; 4Department of Computer Science, Virginia Commonwealth University, Richmond, VA 23284, USA; ranap@vcu.edu (P.R.); pghosh@vcu.edu (P.G.); 5Institute of Biological Science, Federal University of Minas Gerais, Belo Horizonte 31270-901, MG, Brazil; acybelle@gmail.com (A.C.P.G.); vascoariston@gmail.com (V.A.)

**Keywords:** *Corynebacterium pseudotuberculosis*, RNA-Seq, co-expression networks, influence genes, stress condition

## Abstract

*Corynebacterium pseudotuberculosis* is a Gram-positive bacterium that causes caseous lymphadenitis, a disease that predominantly affects sheep, goat, cattle, buffalo, and horses, but has also been recognized in other animals. This bacterium generates a severe economic impact on countries producing meat. Gene expression studies using RNA-Seq are one of the most commonly used techniques to perform transcriptional experiments. Computational analysis of such data through reverse-engineering algorithms leads to a better understanding of the genome-wide complexity of gene interactomes, enabling the identification of genes having the most significant functions inferred by the activated stress response pathways. In this study, we identified the influential or causal genes from four RNA-Seq datasets from different stress conditions (high iron, low iron, acid, osmosis, and PH) in *C. pseudotuberculosis*, using a consensus-based network inference algorithm called miRsigand next identified the causal genes in the network using the miRinfluence tool, which is based on the influence diffusion model. We found that over 50% of the genes identified as influential had some essential cellular functions in the genomes. In the strains analyzed, most of the causal genes had crucial roles or participated in processes associated with the response to extracellular stresses, pathogenicity, membrane components, and essential genes. This research brings new insight into the understanding of virulence and infection by *C. pseudotuberculosis*.

## 1. Introduction

In the past few years, several genomic, transcriptomic, and proteomic studies have been performed to understand the biological basis of *Corynebacterium pseudotuberculosis*; these studies allowed the identification and understanding of the genomic mechanisms that contribute to the virulence and infection processes used by the bacteria [1,2,3,4,5,6,7]. *C. pseudotuberculosis* is a Gram-positive intracellular bacteria and the etiologic agent of caseous lymphadenitis, a chronic, pyogenic, and contagious disease that affects small ruminants causing considerable economic losses for the farmers and meat industries in many countries [5,8]. The gene expression studies, using RNA-Seq under different conditions, can explain which genes inside the genome are responsible for the bacterial maintenance in precarious and restricted situations in order to prevent bacterial infection and propagation [2,3]. These studies generated a significant amount of data from the bacteria, and such data can help perform new bioinformatics studies to understand the genome complexity and the relationship between the genes using network inference algorithms.

Co-expression network inference is a popular computational domain where several algorithms have been developed for predicting genetic interactions. These models are built using statistical techniques on high-throughput experimental gene expression data (such as RNA-Seq and/or microarrays) [9,10,11,12,13]. These models allow the construction of co-expression networks (CEN) to describe the correlation and interactions between the transcriptional genes in the organism. This type of network allows for the understanding of the regulatory mechanisms and processes in the biological system [11,13]. The CEN shows the relationship between genes and the regulatory processes by following the central dogma of regulatory control; network analysis on the CEN can identify significant genes possessing stronger influence or causality in the network [11,13,14].

The influential genes are the minimal set of causal genes (seed nodes) in the network that, when perturbed initially, leads to influence diffusion to other nodes and finally impacts the maximal number of genes in the network. This concept is based on the popular influence maximization algorithms from the social network domain. The activated seed genes can spread their influence by probabilistically activating their neighboring genes based on their expression levels and the edge weights in the CEN. Such influential genes could play a crucial role in the regulation of the gene expression process under specific conditions, such as stress [15,16].

In this study, we identified the influential or causal genes using four RNA-Seq datasets from *C.pseudotuberculosis*, first by using the miRsigpipeline to obtain the predicted gene coexpression network [17] and next applying the miRinfluence tool to identify the influential and causal genes inside the network [15]. We adapted the methodology in these tools to determine the critical genes that play a causal role in the signaling cascade through influence diffusion and hence may regulate the overall gene expression of the entire network.

## 2. Methodology

### 2.1. Bacterial Strains and Growth Conditions

In this study, we used four strains of *Corynebacterium pseudotuberculosis*, isolated from different animals. *C. pseudotuberculosis* 1002 (CP-1002) was a wild strain belonging to biovar ovis, isolated from a caprine host in Brazil [18]. *C. pseudotuberculosis* 258 (CP-258), biovar Equi, was isolated from a horse in Belgium [19]. The variation CP-T1 was a pathogenic wild-type belonging to biovar ovis, isolated from a caseous granuloma found in CLA-affected goats in Brazil [20]. the strain CP-13 was an iron-acquisition-deficient mutant and was generated by [20], employing the in vivo insertional mutagenesis of the reporter transposon-based system TnFuZ in the strain T1. The molecular characterization of the Cp13 mutant showed that the insertion disrupted the ciuA gene, which encodes a putative-iron transport binding protein of the ciuABCD operon [21,22].

The strains of CP-1002 and CP-258 were subjected to different stress levels (acid, osmotic, and heat). The bacteria were grown in Petri dishes containing brain heart infusion (BHI) media. For the acid stress condition, the media was supplemented with hydrochloric acid (which changed the pH to 5); osmotic stress was achieved with 2 M NaCl; thermal stress was induced by pre-heated BHI medium to 50 ${}^{\circ }C$; and in the control condition, the bacteria were grown in BHI medium at physiological condition [2,3,19].

The strains CP-T1 and CP-13 were grown individually either in the presence of the iron chelator $2,2{}^{\prime }$-dipyridyl (DIP) (low iron condition) or without it (high iron condition). The iron-chelated BHI medium was prepared with 250 μM of ferrous iron chelator $2,2{}^{\prime }$-dipyridyl (Sigma Aldrich), which due to its low aqueous solubility, was prepared with 40% (v/v) of ethanol (0.5 M $2,2{}^{\prime }$-dipyridyl stock solution) [22].

### 2.2. Expression Datasets

The cDNA samples from CP-258 and CP-1002 were used to prepare eight individual single-end libraries that were sequenced using the SOLiD™ 3 Plus system platform, to produce 50-nucleotide RNA reads [3]. The datasets used in this study were obtained from the ArrayExpress repository with the accession numbers E-MTAB-2017 and E-MTAB-9217.

From CP-T1 and CP-13, the cDNA samples were used to prepare 14 individual single-end libraries, with three replicates per stress condition, to produce fragments with an average size of 100–200 nucleotides. The Ion Proton Platform was used to perform two rounds of sequencing [22]. The sequencing data were obtained from the Gene Expression Omnibus (GEO) repository with the accession number GSE114125.

Raw data quality was examined using the FastQC tool v0.11 [23]. Per-base quality filtering was performed with Trimmomatic v0.39 [24] with a sliding window trimming approach, with the following parameters: LEADING:3 TRAILING:3 SLIDINGWINDOW:4:14 MINLEN:30 for CP-258 and CP-1002; and LEADING:3 TRAILING:3 SLIDINGWINDOW:4:15 MINLEN:36 for CP-13 and CP-T1. The adapters in CP258 and CP-1002 were removed using Cutadapt [25] using the SOLID Small RNA Adapter sequences.

The RNA-Seq expression data were quantified and aligned using the Salmon [26] tool, where the genomes CP-13 (NZ_CP014998), CP-258 (NC_017945.3), CP-T1 (NZ_CP015100.2), and CP-1002B (NC_017300.2) were used as references. The RNA-Seq expression profile was normalized by transcripts per kilobase million (TPM) [27]. Table 1 shows a summary of the replicon information from NCBI [28]. Table 2 presents the average nucleotide identity (ANI), which signifies the nucleotide-level genomic similarity in the coding regions between the *C. pseudotuberculosis* strains, calculated using pairwise ANI in [29].

The whole genome gene (WEG) expression datasets were selected from all the expressed genes in the differential expression tests. An analysis of differential expression was performed using the GFOLD tool [30] for CP-258 and CP-1002 data and edgeR [31] for CP-T1 and CP-13 data to generate the differentially expressed gene (DEG) datasets. To select the DEG, we used a fold-change of 2 and a *p*-value <0.05 [32].

### 2.3. Network Analysis

The predicted gene interaction networks (or equivalently termed as gene coexpression networks in some cases) were built with miRsig [17]. This tool applies seven different network inference algorithms on the gene expression data that individually reverse-engineer the interaction scores between the genes. Next, a consensus-based approach was applied in miRsig to estimate the overall score of every gene-gene interaction using an average ranking based approach to infer finally the gene coexpression network with scores on edges depicting the likelihood of possible genetic interactions between them. This algorithm was adopted in this work to perform the network analyses with the gene expression data.

For each strain, we inferred two gene coexpression networks: (I) with the whole genome expression and (II) only with the differentially expressed genes in all the stressed conditions.

For the consensus ranking scores, we selected 0.85 as the cut-off for the all expressed genes dataset and the differentially expressed genes datasets; these cut-offs ensured high confidence on the edge scores on gene coexpression networks on which the following causal gene identification methods were applied [15].

### 2.4. Identification of Causal Genes

To identify the causal genes inside the network, we used the miRinfluence algorithm [15]; this algorithm uses network diffusion theory to quantify the influence of individual genes on the signal transaction process in the gene interaction networks across different conditions or stress. The algorithm ranks all the genes in the network according to their influence scores after calculating the optimal coverage (in terms of the number of nodes influenced in the network) with 10,000 Monte Carlo simulation based random walks.

We selected the top twenty causal genes from the whole genome expression networks while we selected the top ten causal genes for the DEG networks. These top genes showed a higher influence score in the influence diffusion model of miRinfluence inside the network.

The final list of causal genes was compared to the database of Online GEne Essentiality (OGEE) [33] using the *Mycobacterium tuberculosis* dataset, which is a phylogenetically close organism to *C. pseudotuberculosis* available in the database and also with the Rocha et al. reference genes’ study [34]. The OGEE classifies genes into three types: essential, nonessential, and a particular condition called conditionally essential genes with variable essentiality statuses across datasets.

### 2.5. Network Clustering

To analyze the gene interaction network’s inherent structure, we performed a cluster analysis using the K-means [35,36] algorithm within ClusterMaker [37] in Cytoscape [38], where the distance between the genes was calculated by the Euclidean distance. We used the same tool to discover the optimal number of clusters through the silhouette metric for each network. K-means was performed to identify the cluster with one or more influential genes present using the betweenness-centrality, degree, and closeness-centrality node attributes as the clustering input.

### 2.6. Sub-Network Detection and Enrichment Analysis

We selected the sub-networks having one or more influential genes present to perform the gene annotation analysis. The enrichment analysis was performed with the GO FEAT platform [39] using the gene influence nucleotide sequences and StringApp v.11.0 [40] in Cytoscape Version 3.8.0 [38] using *Corynebacterium pseudotuberculosis* as reference species, with >0.70 for the confidence cut-off and <10 for the maximum additional interactors’ score. The enrichment of the clusters’ metabolic pathways was performed using the clusterProfiler R package using the enrichKEG with CP-1002 (KEGG ID: cpk) and CP-258 (KEGG ID: coe) as reference organisms [41,42]. Visual representation was created using ggplot2 v.3.3.0 [43] and ggpubr v.0.3.0 [44].

## 3. Results

Read quality assessment was done through FastQC v0.11. Samples CP-1002 and CP-258 presented a Phred score distribution among 16–29, while CP-13 and CP-T1 produced Phred scores between 20 and 26. Base quality trimming was performed with Trimmomatic; for samples of the CP-T1 and CP-13 strains, this procedure excluded 1% and 5% of the total reads, respectively; for samples CP-259 and CP-1002, eight to 10% of total reads were excluded from each one. Adapter filtering was performed with CutAdapt in CP-258 and CP-1002. Trimmed datasets produced by this method were mapped to CP-1002B (NZ_CP012837), CP-T1 (NZ_CP015100.2), CP-258 (NC_017945), and CP-13 (NZ_CP014998) reference genomes. For CP-258, the percentage of mapped reads to the reference genome ranged from 57% to 67%; CP-1002 presented mapped read percentages ranging from 58% to 72.68%; CP-13 strain mapping covered between 63.48% and 90.59% of reads; and CP-T1 mapped between 54.55% and 67.60% of trimmed reads (Table 3).

Differential expression analyses were performed with edgeR [31] on the CP-13 and CP-T1 samples. A total of 93 genes for CP-T1 and 62 genes for CP-13 were found to be differentially expressed between control and high iron conditions. We utilized GFold [30] to perform differential analyses in CP-258 and CP-1002; this method yielded 167 differentially expressed genes in CP-1002 and 138 genes in CP-258 (Table 3).

Using the miRsig tool, we built gene coexpression networks with four datasets, two for each *C. pseudotuberculosis* strain; Table 1 shows the sizes of these datasets. For the whole genome expressed datasets, we predicted a network with 86,367 gene-gene interactions for Cp-13, 9376 interactions for CP-258, 6682 for CP-1002, and 107,202 for CP-T1 (Figure 1). For the differentially expressed networks, we predicted a total of 46 gene-gene interactions for CP-13, 165 interactions for CP-258, 155 for CP-1002, and 98 for CP-T1. The whole genome and differentially expressed genes networks are provided in the Appendix A.

We selected the influential genes using the coexpression network as the input to the miRinfluence tool. From the output set of ranked genes based on their respective influence scores, we selected the top 20 for the whole genome expressed network and the top 10 for the DEG network. Table 4 and Table 5 show the gene annotation for each network. Seventy-five percent of these genes showed a high degree distribution compared with the average of the other nodes in the network.

From the causal genes’ list obtained from the whole genome expressed datasets, we found that 15% of the genes in Cp-258, 25% in CP-13, 10% in CP-T1, and 25% in Cp-1002 were considered as essential and conditionally essential genes in *Mycobacterium tuberculosis* according to the OGEE [33,34]. This implied that these genes may be involved in important functions within the strains of the bacteria that were studied.

In the whole expressed genome list, the genes *galU* and *argS* were categorized as essential, and gene rmlD was conditionally essential in strain CP-258. In CP-T1, the genes *pdpB* and *trpC* were classified as essential genes. In strain Cp-1002, the genes *uvrD3*, *whiB*, *rplO*, *udgA*, and *uvrA* were conditionally essential genes. *serC*, *mraY*, and *glmS* were listed as essential, and the genes *sdaA* and *lpdA* were classified as conditionally essential genes according to [33,34].

For the DEG list, the gene metX in CP-13, dnaK in CP-1002, and lysA2 in CP-T1 were labeled as essential genes. cdd in CP-1002 and cstA in CP-258 were classified as conditionality essential according to [33,34].

We made the functional annotation using GO FEAT [39] and Cytoscape StringApp [40]; these tools allowed the characterization and functional annotation of the causal gene groups present in each of the studied genomes in the whole genome expression and differential genome expression datasets.

Figure 2 shows the pathways of the genes in the whole expressed genome datasets. The pathways with more genes in CP-13 were the biosynthesis of antibiotics with 13 genes and biosynthesis of amino acids with nine genes; the nucleotide excision repair and pyrimidine metabolism were the more active pathways with seven genes and five genes respectively in CP-1002; for CP-258, the two-component system pathway with four genes and propanoate metabolism pathway with three genes were activated. In CP-T1, the ABC transporter was the more activated pathway.

Figure 3 shows the KEGG pathways found in the DEG datasets. We found 11 pathways in four *C. pseudotuberculosis* strains; the pathway with the most genes related to porphyrin and chlorophyll metabolism, expressed in CP-1002 and CP-T1. Other important pathways were metabolic pathways, fatty acid metabolism, citrate cycle (TCA cycle), biosynthesis of unsaturated fatty acids, and biosynthesis of amino acids; these pathways are involved in the cell walls’ components and protect the bacteria from environmental stress [45].

Appendix A, shows the results of the gene ontology analysis for the top 20 causal genes. The cell adhesion, SOS response, cell division, carbohydrate metabolism process, transmembrane transport, methylation, and regulation of transcription-DNA-templated were the biological processes in which the most genes participated in the four studied strains. Concerning cellular components, most of the causal genes were part of the cytoplasm, integral component membrane, and plasma membrane in the four strains. The majority of causal genes in these genomes participated in molecular functions such as ATP binding, ATPase activity, metal ion binding, DNA, and transferase activity.

Considering the top 10 causal genes identified in the differentially expressed datasets (Figure 4), most of these genes participated in molecular functions such as iron-sulfur clusters’ binding, transmembrane transport activity, DNA binding, transferase activity, and oxidoreductase activity. In the cellular components’ ontology, these genes were part of the integral components of the membrane and the cytoplasm. Considering the biological processes, the top 10 differentially expressed genes related to processes such as cell adhesion, cellular iron ion homeostasis, the nitrogen compound transport nucleoside metabolic process, the phosphorelay signal transduction system, and other processes.

We performed the K-means algorithm in all the coexpression networks. For whole expressed genomes in Cp-13, the silhouette metric predicted 24 clusters, and the causal genes were present in 12 of these clusters. In Cp-258, the metric identified 21 clusters with all the causal genes present in four clusters. In the CP-1002 strain, it identified 16 clusters, and the casual genes were distributed into four clusters. Finally, in CP-T1, we identified 30 clusters, and the causal genes were assigned to five clusters (Appendix A).

In the differentially expressed genes’ network for CP-258, the silhouette metric split the network into six clusters, and the causal genes were distributed into two clusters. For CP-1002, we detected six clusters, and all the influential genes were represented in four clusters. In the CP-13 strain, the network was divided into four clusters, and the causal genes were present in three clusters. Finally, the network of CP-T1 was split into five clusters, and the causal genes were assigned to three clusters (Appendix A).

In the clusters with causal genes, we performed gene enrichment analysis using the clusterProfiler package to identify the KEGG pathways in these clusters. Figure 5 shows the pathways involved in the clusters from the whole expressed genes network in CP-13 and CP-1002; the other clusters’ figures are in Appendix A. The more representative pathways in the clusters in all the strains were metabolic pathways and biosynthesis of secondary metabolites, which were present in almost all the clusters of the studied genomes. Other interesting pathways activated by the causal genes in these clusters were biosynthesis of amino acids, microbial metabolism in diverse environments, quorum sensing, ribosome, and carbon metabolism metabolites.

For the top 10 influential genes from the differentially expressed genes’ networks, we found that the metabolic pathway was the more activated one in all strains. Other pathways with more representation in the strains were biosynthesis of antibiotics, microbial metabolism in diverse environments, biosynthesis of secondary metabolites, ABC transporters, and carbon metabolism. The figures are in Appendix A.

## 4. Discussion

Intracellular pathogens have mechanisms of response to conditions of harmful extracellular stresses caused by the host. Among the types of stresses that a bacterium can face are the drastic change in temperature, pH alteration, sudden changes in osmolarity, and the presence of reactive oxygen species, among others. In this study, we identified the influential genes referring to the stresses mentioned above from experiments performed by [3,22]. In the strains analyzed, the genes that controlled most interactions within the co-expression network were related to the response to extracellular stresses, pathogenicity, membrane components, and essential genes.

The stress conditions faced by *C. pseudotuberculosis* during the infectious process are diverse, from entry into the host, through the lymphatic system, to intracellular replication in macrophages, and the establishment of lesions within the organs [46]. The strains CP-13 (mutant) and CP-T1 (wild) were subjected to restriction of iron, an essential micronutrient for the proliferation of pathogens. In the strains CP-1002 and CP-258, three types of stresses were applied: acidic, thermal, and osmotic. These conditions simulated the environment found by the bacteria during infection in the host.

Between the top 20 causal genes of the CP-13, a mutant with the disrupted ciuA gene that encodes a putative iron transport binding protein from the ciuABCD operon may be involved in the stress response. These genes are *mca*, *Hisn*, *rbsK*, *pepC2*, *tcsR3*, *MRPD*, *ALD*, *LPDA*, *COPD*, and Cp13_RS10225 Cp13_RS07170 (Table 4). The products of these genes showed a relationship with reactive oxygen species (ROS). ROS can cause oxidative stress, which is a condition of imbalance between ROS production and its removal through systems (enzymatic or non-enzymatic) that remove or repair the damage caused by them [47]. Bacteria are regularly exposed to free radicals present in the extracellular environment.

Actinobacteria, the phylum that *C. pseudotuberculosis* is included in, are capable of resisting extracellular ROSs present in the phagocytic environment, produced by macrophages against pathogen invasion [48]. In response to ROSs, neutralization mechanisms, called antioxidants, are used. For example, the mca gene is involved in the mycothiol metabolic process (Figure 4). This compound is believed to act as an antioxidant like glutathione, keeping the intracellular medium free of alkylating agents and other toxins in Gram-positive bacteria [49]. The mycothiol loss in Mycobacteria is associated with slow growth and increased sensitivity to reactive oxygen species and antibiotics [50]. Moreover, the *hisN* gene encodes a protein involved in histidine biosynthesis, which in Actinobacteria is converted into the antioxidant ergothioneine [51,52].

In the differential expression analysis, the genes Cp13_RS02610 and Cp13_RS09955 were repressed in the iron limitation condition (Appendix A). They are involved in redox processes, in which they usually occur during cellular respiration or in oxidative stress conditions producing free radicals [47]. These two genes could be participating in the production of ROS, but they may have been repressed with the expression of antioxidants. Other genes identified in CP-13 encode proteins related to cell adhesion.

The T1 wild-type strain also showed genes that may be involved in the stress response, such as the *ogt*, Cp13_RS09405, and Cp13_RS07860 genes. The Cp13_RS09405 and Cp13_RS07860 genes are related to redox reactions. The *ogt* gene is involved in the DNA methylation and repair process (Figure 4). DNA repair pathways are essential to maintain the integrity and stability of the genome. Pathogenic bacteria are constantly under external pressure, from the environment and from the interaction with the host, which can cause damage to bacterial DNA [53]. Consequently, the presence of this gene among the influential ones in the co-expression network determines its importance in the face of possible stresses that can destabilize the bacterium’s genome, impairing its survival.

In the differential expression analysis of the CP-T1 strain, the Cp13_RS02285 gene (uncharacterized protein) was induced six times. It is an integral membrane protein. Furthermore, the *hmuT* and *sprT* genes were induced three times (Appendix A). The *hmuT* gene encodes a periplasmic protein that plays a role in the acquisition and transport of hemin. This gene was identified as an influencer and corroborated the results found by [22], who found these genes to be involved in the transport of hemin.

The strains CP-1002 and CP-258 showed causal genes related to stresses *ureE*, *uvrA* (CP-1002), and *tcsS4, mprA_2* (CP-258). Urease (ure) has a route in which bacteria try to alkalize their environment during acidic stress. The neutralization of acids results from the production of ammonia (NH${}_{3}$), which combines with a proton from the cytoplasm to produce ammonium (NH${}_{4}^{+}$), thus raising the internal pH [54]. The urease enzyme promotes the hydrolysis of urea, which acts as an H${}^{+}$ ion receptor, generating a neutral pH inside the bacteria, which gives, for example, *H. pylori* resistance to gastric acidity. Most of the urease synthesized by the bacterium is located in its cytoplasm [55]. The *uvrA* gene also has an adaptive response to acidity [56].

The *tcsS4* and *mprA_2* genes encode an osmosensor kinase and a two-component system response standard (Table 4). Two-component systems may be involved in response to osmotic stress. Osmosensors regulate the expression of genes that encode osmoregulators, constituting two-component systems: the sensor located on the membrane has a histidine kinase domain that in the presence of the stimulus, transmits the information via phosphorylation to the response regulators. In *E. coli*, the EnvZ-OmpR system, which regulates the expression of the *OmpC* and *OmpF* porins, facilitates the diffusion of hydrophilic molecules. In response to an increase in osmotic pressure, the expression of *OmpF* is decreased, and *OmpC* has its expression increased [57].

In addition, coders were identified in relation to cell adhesion: the Cp1002B_RS02920 (CP-1002), Cp1002B_RS10840, and CP258_RS03105 (CP-258) genes. These genes encode adhesive proteins that are important for binding the receptor to host cells during pathogenesis. For example, *Escherichia coli PapG* adhesin from pilus P is necessary for binding to the human renal receptor during the pathogenesis of pyelonephritis [58].

In the analysis of differential expression, the Cp1002B_RS02920 gene was repressed in osmotic stress (Appendix A). The *vapI* gene (CP-258) encodes a protein associated with virulence (induced in osmotic stress). Furthermore, the *rshA* gene (CP-258) produces an anti-sigma factor and contains a CXXC motif like a thiol-disulfide redox switch [59]. This gene was induced in acid stress (Appendix A), confirming that this condition is essential for the analysis of the genes of *C. pseudotuberculosis* that may be involved in the host’s infection, mainly related to the infection of phagocytic cells.

The biosynthesis of the antibiotic pathway, which is regulated by phosphate (Figure 2), is activated by the environment’s nutritional stress such as carbon or nitrogen limitation. The antibiotics produced for this pathway can destroy or inhibit the growth of other bacteria in the microfilms in the nutrients’ competition; furthermore, some antibiotics may have inter-cellular communication functions in the communities [60,61]. Other critical pathways are microbial metabolism in a diverse environment related to the stress response [45]; the biosynthesis of amino acids pathways are connected with central carbon, nitrogen, and sulfur metabolism [62], and nucleotide excision repair can help repair DNA that has been damaged by different stresses [63].

Through these pathways were identified by the gene clusters in the whole genome expression and differentially expressed gene datasets, we could infer that the causal genes identified in the network allowed the bacteria to respond to the stress’ stimulus, activating and co-regulating the expression of their neighbors. These influential genes could send some alert signals to activate the regulatory factors in these pathways to stimulate or inhibit the translation, transcription, or the expression of other genes, to generate a physiological and biochemical adaptation in response to the environmental stress [3]. An essential point in the network analysis is the highest degree rates represented for the clusters with more influential genes, which could indicate their regulatory role in controlling their other neighbors, whose expressions are correlated in the network [14].

The influential genes identified in the differential expression datasets were correlated with basal cellular processes inside the genome. Moreover, we identified pathways using KEGG [42], and essential genes were compared in our results with the dataset of *Mycobacterium tuberculosis* from OGEE [33], which were related to the defense and adaptation of bacteria to different stresses; this could mean that these influential genes play a critical role in the synthesis of proteins and the survival process of the bacteria under the different stress conditions. It is important to highlight that bench-top experiments are required to validate these genes as essential in *C. pseudotuberculosis*.

## 5. Conclusions

In this work, we developed a co-expression network analysis to identify the ranked influential and causal gene sets using the gene expression datasets from four strains of *C. pseudotuberculosis* under different stress conditions. The network analyses were performed using computational methods based on the information diffusion concept to identify these genes. For the analysis, we used both cases of considering all the expressed genes and only the deferentially expressed genes to compare the detected genes in both networks. These causal genes were shown to play a critical role in activating other genes to generate the bacterial response against the stress conditions in the environment.

## Figures and Tables

**Figure 1 genes-11-00794-f001:**
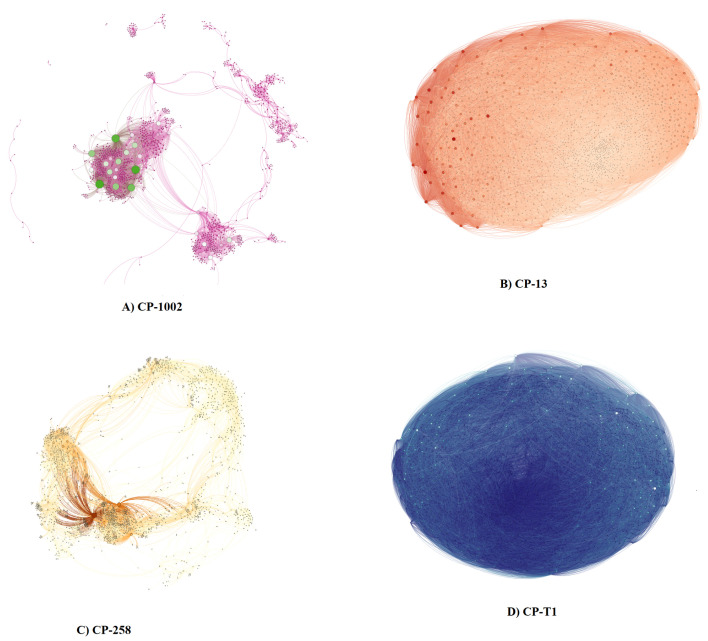
Whole expressed gene dataset networks. (**A**) The network generated by the data isolated from *C. pseudotuberculosis 1002*. (**B**) The network generated by the data from *C. pseudotuberculosis strain CP-13*. (**C**) The network generated by the data from C. pseudotuberculosis 258. (**D**) The network generated by the data from *C. pseudotuberculosis* strain T1.

**Figure 2 genes-11-00794-f002:**
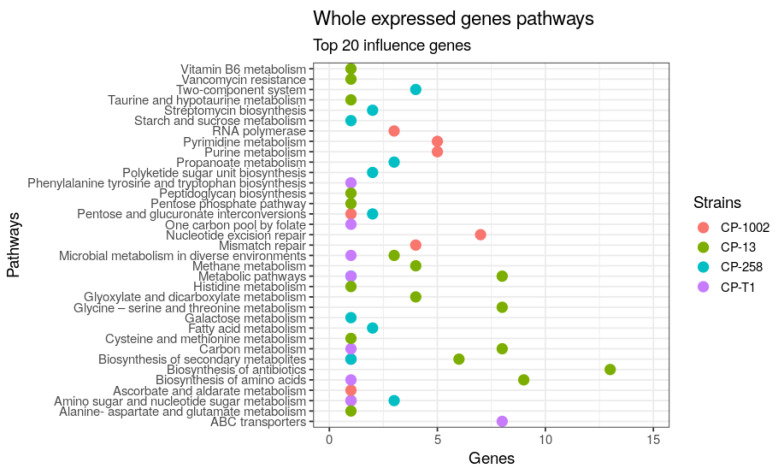
The pathways that involve the top 20 genes identified in the whole expressed genome network in CP-258, CP-13, CP-T1, and CP-1002.

**Figure 3 genes-11-00794-f003:**
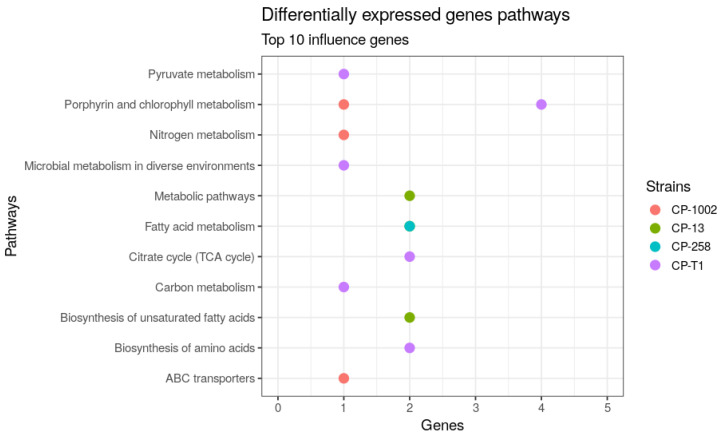
Pathways involved with the top 10 genes identified in the differential expression network in Cp-258 and Cp-1002.

**Figure 4 genes-11-00794-f004:**
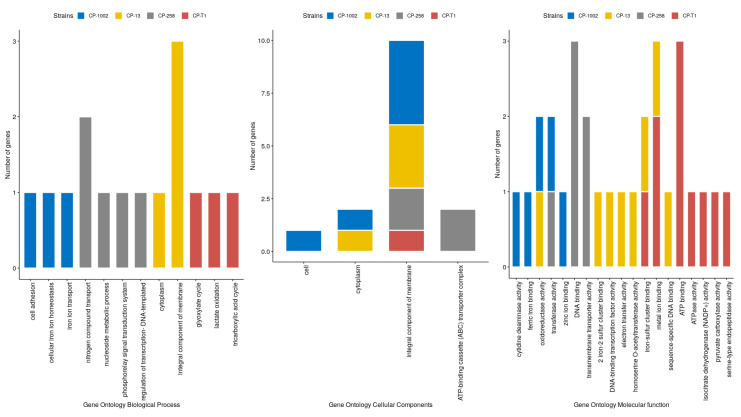
Gene Ontology results of differentially expressed genes in CP-1002, CP-258, CP-13, and CP-T1. Left: biological process; center: cellular components; and right: molecular function.

**Figure 5 genes-11-00794-f005:**
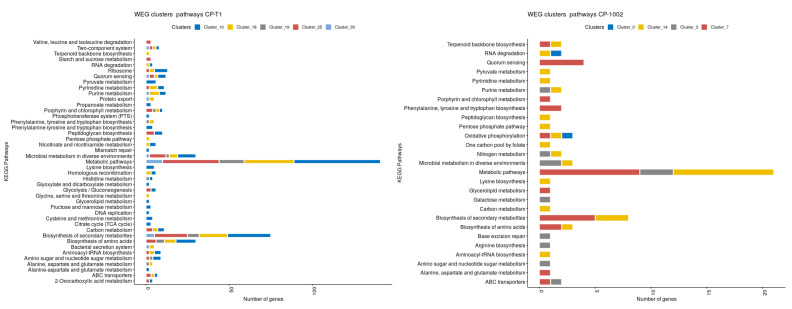
Pathways of the clusters where the influential genes were present in the whole expressed genome network. Left: CP-1002; right: CP-T1.

**Table 1 genes-11-00794-t001:** Replicon information summary of each *C. pseudotuberculosis* strain.

Strains	Size (Mb)	GC%	Protein	Genes
**CP-13**	2.34	52.2	2013	2135
**CP-T1**	2.34	52.2	2008	2125
**CP-258**	2.37	52.1	2038	2165
**CP-1002B**	2.34	52.2	2009	2124

**Table 2 genes-11-00794-t002:** Average nucleotide identity (ANI) similarity between the *C. pseudotuberculosis* strains.

Strains 1	Strains 2	ANI%
**CP-13**	**CP-258**	98.9036
**CP-13**	**CP-1002B**	99.9966
**CP-1002B**	**CP-258**	98.9051
**CP-1002B**	**CP-T1**	99.9968
**CP-T1**	**CP-258**	98.904
**CP-T1**	**CP-13**	100

**Table 3 genes-11-00794-t003:** Sizes of the gene expression datasets.

Strains	CP-13	CP-258	CP-1002	CP-T1
**Whole Genomes**	2113	2064	2091	2093
**Differentially Expressed Genes**	63	139	168	93

**Table 4 genes-11-00794-t004:** Products generated by the top 20 influential genes identified in the networks built with the whole genome expression data from Cp-13, Cp-258, and Cp-1002.

Genes Cp-13	Product	Genes Cp-258	Product	Genes Cp-1002	Product	Genes Cp-T1	Product
***mca***	Mycothiol S-conjugate amidase	**CP258_RS02980**	Methylmalonyl-CoA carboxyltransferase 12S subunit	**Cp1002B_RS03590**	Hypothetical protein	***pknL***	Serine/threonine protein kinase
***hisN***	Histidinol-phosphatase	**CP258_RS03105**	LPxTG domain-containing protein	***mcbR***	TetR family transcriptional regulator	***fhs***	Formate–tetrahydrofolate ligase
***rbsK***	Ribokinase	**CP258_RS03110**	Aminotransferase	***ureE***	Urease accessory protein UreE	***ogt***	Methylated-DNA–protein-cysteine methyltransferase
***pepC2***	M18 family aminopeptidase	***tcsS4***	Two component system sensor kinase protein	***uvrD3***	DNA helicase	**Cp13_RS09405**	Oxidoreductase
***tcsR3***	Two-component system transcriptional regulatory protein	**CP258_RS04420**	Hypothetical protein	**Cp1002B_RS02755**	Abi family protein	***pbpB***	Penicillin binding protein transpeptidase
**Cp13_RS10225**	Antimicrobial peptide ABC transporter ATPase	***rimJ***	Ribosomal-protein-alanine acetyltransferase	***cmtB***	Trehalose corynomycolyl transferase B	**nodI**	Nod factor export ATP- binding protein I
***ald***	Alanine dehydrogenase	**CP258_RS03575**	ABC transporter ATP-binding protein	***whiB***	Transcriptional regulator WhiB	**Cp13_RS04005**	Hypothetical protein
***mrpD***	Na(+)/H(+) antiporter subunit D	***tehA***	C4-dicarboxylate transporter/malic acid transport protein	***rplO***	50S ribosomal protein L15	***pafA2***	Pup–protein ligase
**serC**	Phosphoserine transaminase	**CP258_RS03580**	Antibiotic biosynthesis monooxygenase	**Cp1002B_RS02920**	LPxTG domain-containing protein	**Cp13_RS01180**	Secreted hydrolase
***mgtA***	Glycosyl transferase group 1	***echA6***	Enoyl-CoA hydratase echA6	***rluC***	Ribosomal pseudouridine synthase	***trpC***	N-(5’-phosphoribosyl)anthranilate isomerase
**Cp13_RS01600**	ABC-type metal ion transport system, periplasmic component/surface adhesin	***rmlD***	dTDP-4-dehydrorhamnose reductase	***yhcL***	Cryptic C4-dicarboxylate membrane transporter dcuD	**Cp13_RS08505**	Acetyltransferase
***sdaA***	L-serine dehydratase	**CP258_RS07545**	Hypothetical protein	**Cp1002B_RS01655**	ABC transporter inner membrane protein	***nagB***	Glucosamine-6-phosphate deaminase
**Cp13_RS07170**	Glyoxalase/bleomycin resistance protein/dioxygenase	***galU***	UTP–glucose-1-phosphate uridylyltransferase	**Cp1002B_RS10840**	LPxTG domain-containing protein	**Cp13_RS03565**	Hypothetical protein
**Cp13_RS01385**	Hypothetical protein	***cstA***	Response regulator	**Cp1002B_RS01705**	Hypothetical protein	**Cp13_RS05420**	ABC transporter ATP-binding protein
***lpdA***	Flavoprotein disulfide reductase	***argS***	Arginine–tRNA ligase	**Cp1002B_RS10170**	ABC transporter	***yvrC***	ABC transporter substrate-binding protein
***mraY***	Phospho-N-acetylmuramoyl pentapeptide-transferase	***hpf***	Ribosome hibernation promoting factor	***fadF***	Protein fadF	***fagA***	Hypothetical protein
**Cp13_RS08655**	Hemolysin III-like protein	***deoA***	Thymidine phosphorylase	***udgA***	UDP-glucose 6-dehydrogenase	***mnmA***	tRNA-specific 2-thiouridylase MnmA
**copD**	Copper resistance D domain- containing protein/Cytochrome c oxidase caa3 assembly factor (Caa3_CtaG)	***mprA_2***	Two component system response regulator	***gltT***	Sodium/glutamate symporter	**Cp13_RS06640**	Hypothetical protein
***glmS***	Glutamine–fructose-6-phosphate aminotransferase [isomerizing]	***oppC2***	Oligopeptide transport system permease OppC	**Cp1002B_RS01200**	Serine proteases of the peptidase family S9A	**Cp13_RS07860**	Phosphoglycerate dehydrogenase
**Cp13_RS10090**	NYN domain-containing protein	***scpB***	Segregation and condensation protein B	***uvrA***	UvrABC system protein A	***Cp13_RS03800***	Putative secreted protein

**Table 5 genes-11-00794-t005:** Proteins produced by the top 10 influential genes identified in the networks built with the differentially expressed genes’ data from Cp-258 and Cp-1002.

Genes Cp-13	Product	Genes Cp-258	Product	Genes Cp-1002	Product	Genes Cp-T1	Product
***desA3***	Stearoyl-CoA 9-desaturase	**CP258_RS02710**	Putative secreted protein	***cynT***	Carbonic anhydrase	***pyc***	Pyruvate carboxylase
***htaA***	Cell-surface hemin receptor	**CP258_RS02905**	Hypothetical protein	**Cp1002B_RS03070**	HtaA domain-containing protein	***fecE***	Fe(3+) dicitrate transport ATP-binding protein FecE
***Cp13_RS02285***	Hypothetical protein	***vapI***	Virulence-associated protein	**Cp1002B_RS02920**	LPxTG domain-containing protein	***hmuT***	Hemin-binding periplasmic protein
***lutB***	Putative iron-sulfur protein	***tetR3***	TetR family transcriptional regulator	***cdd***	Cytidine deaminase	**Cp13_RS04105**	LUD_dom domain-containing protein
**Cp13_RS04105**	LUD_dom domain-containing protein	***rshA***	Anti-sigma factor	**Cp1002B_RS03075**	Hypothetical protein	**Cp13_RS02285**	Hypothetical protein
***htaB***	Cell-surface hemin receptor	***cstA***	Response regulator	**Cp1002B_RS03455**	Oxidoreductase	***lysA2***	Diaminopimelate decarboxylase
**Cp13_RS02610**	Stearoyl-CoA 9-desaturase electron transfer partner	**gluC**	Glutamate transport system permease protein gluC	**Cp1002B_RS03180**	Hypothetical protein	***icd***	Isocitrate dehydrogenase [NADP]
**Cp13_RS09955**	Flavin reductase	***odhI***	Oxoglutarate dehydrogenase inhibitor	***ftn***	Ferritin	***sprT***	Trypsin
***metX***	Homoserine O-acetyltransferase	**yecS**	ABC transporter domain-containing protein	**dnaK**	Chaperone protein DnaK	***gluA***	Glutamine ABC transporter ATP-binding protein
***ripA_2***	HTH-type transcriptional repressor of iron protein A	***ykoE***	HMP/thiamine permease protein ykoE	**glpQ_1**	Glycerophosphoryl diester phosphodiesterase	***lutB***	Putative iron-sulfur protein

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
