# Peer review of "Co-Expression Networks for Causal Gene Identification Based on RNA-Seq Data of Corynebacterium pseudotuberculosis"

_genes, 2020, doi:10.3390/genes11070794_

Round 1

Reviewer 1 Report

All my points from the last review were addressed accordingly. 

I have no further comments.

Author Response

We would like to thank the reviewer for their insightful comments on the paper, which significantly improved it.

Reviewer 2 Report

There has been a misunderstanding related to the correction methods that authors need to fix it.

TPM is also a genome-wide correction which works exactly in the same way as RPKM.

Genome-wide correction methods are good in removing inter-replicate biases. However, the bias for total cellular RNA is an inter-sample error. Briefly, we need to compare same number of cells as functional unit of living organisms among samples/treatments.

In practice, we use same quantity of RNA and try to remove biases among replicates using genome-wide correction like TPM or RPKM. The disadvantage of these methods is that they cannot remove the bias for difference in number of cells. For example, if treatment changes total cellular RNA content by 25%. This will lead in 25% variation in cell number between the treated and untreated samples when we use same quantity of RNA for the experiment. That means comparing for example 100 cells vs 125 cells. This bias cannot be removed by genome-wide correction methods.

To exclude both of the mentioned bias sources, a combinatorial approach, i.e., a genome-wide correction plus reference gene based approach is needed.   

My advice for the first revision was as:

“RPKM is a genome-wide correction method. This assumes no changes in total cellular RNA. Under some conditions however total RNA changes which makes genome-wide corrections not suitable method. When there is no total cellular RNA data, therefore, it is recommended to apply a combinatorial approach, i.e., genome-wide and ref. gene based approaches, (see Journal of Biotechnology 188 (2014) 100–109 https://doi.org/10.1016/j.jbiotec.2014.08.012) to ensure expelling all kind of biases.”

Author Response: As suggested by the reviewer, we reviewed our methodology and decided to change the normalization method to Transcripts Per Kilobase Million (TPM).

Other major comments:

A) Expression fluctuations are two types;

  1. The treatment directly regulates and changes the expression of genes. This is not a choice of organism. This we can call first wave or primary response.
  2. The secondary response which the organism gives to adapt itself.

The first wave expression changes are not necessarily beneficial since the evolution is an ongoing process. Therefore, it is not possible at least for majority of genes to define their expression changes and it is a safe approach to call all as found expressional changes. So, the recommendation is not to use statements like causal genes or genes identified as influential. Otherwise, authors will need to explain how they make sure that a gene with many interaction partners that show a first wave and non-beneficial response (expression change as side-effect) is not considered as causal gene.

B) To call a gene as essential, it should be lethal when it is knocked out. So, authors need amendments accordingly either by providing references to the essentiality of genes or revise their manuscript.

Reviewer 3 Report

The authors have addressed the comments, and the manuscript has been sufficiently improved. It can now be published in its present form after correction of several typos (e.g., on line 199, and in the title - it apparently should be read as 'based on RNA-seq data').

Author Response

We would like to thank the reviewer for their insightful comments on the paper, as these comments led us to improve it. We took into consideration all reviewer comments and addressed them in the paper.

It can now be published in its present form after correction of several typos (e.g., on line 199, and in the title - it apparently should be read as 'based on RNA-seq data').

Response to the reviewer: Thank you so much for catching this typo error, which we have now corrected.

Reviewer 4 Report

This revised version of the manuscript improved in terms of addressed my prior comments, and now I am pleased to recommend this revised manuscript for publication in Genes.

Author Response

(The authors gave the same response as above.)

Round 2

Reviewer 2 Report

After two rounds of revision, authors have neither provided a logical explanation nor proper data correction for the expression data analysis. Therefore, I have no choice of rejection due to the importance of this on the content of DEGs.

To clarify the authors’ speculations in the cover letter, here I add brief explanation:

Authors claim that the cell number bias has been removed during the sample collection step and by colony count.

First of all, colony count cannot remove this bias for cell number at all. Second, we need to correct for this bias even though we count and use the same number of cells; consider the simplest example where you extract DNA from 1000 cells of a given E.coli strain. You will find up to 2-3 fold changes or even more differences in quantities of the extracted DNAs among the replicates. This indicates that the extracted DNAs do not represent same number of E.coli cells.

This is the same for biases in the quantity of DNA/RNA/protein even though we measure and use same quantities. This is because measurements are not very precise.

So, that is why we always do the correction on data and before analyses. Therefore, regardless of what have been done to minimize the errors, there are some biases left and data correction is always necessary. This is why you have done TPM correction. TPM correction removes the biases for quantity of the used RNA (equal quantity) among the replicates and, increases the efficiency of statistical analysis in order to detect real changes even though they are very small between two treatments/samples.

However, for being able to perform a comparison between the treatments, and before thinking about the efficiency of statistical analysis and the genome-wide corrections (like TPM), we need to have the data from treatments representing same number of cells between the treatments. This can be done by ref. genes based correction as informed by comments in previous reviews.

Another point is that authors have not provided a scientific explanation and reasons for how they should call the DEGs.

These two methodological issues that deal with the focus of the study call for a major revision before being able to go for a deep review and focus on the actual findings and discussions. Based on this, my recommendation is unfortunately rejection.

The last but also important comment is that accepting or rejecting any approach should be by providing scientific reasons. Recalling different published papers and saying that many do a given approach, is not a scientific explanation for the reliability of a method. This is why we do research otherwise many are doing the same things at the moment and many of these actions are not right thing to do. So, we should not forget scientific explanations and reasons.